# Temperature-Controlled Assembly/Reassembly of Two Dicarboxylate-Based Three-Dimensional Co(II) Coordination Polymers with an Antiferromagnetic Metallic Layer and a Ferromagnetic Metallic Chain

**DOI:** 10.3390/polym11050795

**Published:** 2019-05-02

**Authors:** Hui-Chen Yu, Chin-Hsuan Lin, Chen-I Yang

**Affiliations:** Department of Chemistry, Tunghai University, Taichung 407, Taiwan; a010406@yahoo.com.tw (H.-C.Y.); az74410a@gmail.com (C.-H.L.)

**Keywords:** assembly/reassembly, coordination polymer, magnetic properties, antiferromagnetic, ferromagnetic

## Abstract

Two new dicarboxylate-based three-dimensional cobalt coordination polymers, [Co(Me_2_mal)(bpe)_0.5_(H_2_O)]*_n_* (**1**) and [Co(Me_2_mal)(bpe)_0.5_]*_n_* (**2**), were synthesized from dimethylmalonic acid (H_2_-Me_2_mal) in temperature-controlled solvothermal reactions. Lower temperatures (60–80 °C) favored the formation of **1**, while higher temperatures (120 °C) favored the production of **2**. Compound **1** is comprised of Co(II) corrugated layers linked by *syn*–*anti* carboxylate bridges from the Me_2_mal^2−^ ligands and pillared through *bis*-monodentate bpe groups. Compound **2** is comprised of a three-dimensional network involving one-dimensional Co–carboxylate chains bonded by antisymmetric µ_4_-Me_2_mal^2−^ ligands and aligned parallel to the [001] direction. The solvothermal retreatment of crystalline samples of **1** in a DMF/H_2_O solvent at 120 °C allowed the structural reassembly, with complete conversion within **2** over 48 h. Magnetic analyses revealed that compound **1** exhibits both spin-orbital coupling and antiferromagnetic interactions through a *syn*–*anti* carboxylate (Me_2_mal^2−^) bridge exchange pathway [Co–Co separation of 5.478 Å] and compound **2** showed a ferromagnetic interaction resulting from the short Co–Co separation (3.150 Å) and the small Co–O–Co bridging angles (98.5° and 95.3°) exchange pathway which was provided by µ_4_-Me_2_mal^2−^ bridging ligand.

## 1. Introduction

Coordination polymers (CPs), hybrid crystalline materials comprised of organic and inorganic components whose structures are extended by coordination bonds, have attracted considerable interest in the field of condensed matter [1,2,3,4,5,6,7,8,9,10]. The beneficial features of these materials are attributed to the building blocks from which they are constructed, which have of both organic and inorganic parts, thus conferring hybrid properties. Consequently, they commonly possess novel and fascinating properties or functionalities, which originate from the hybridization of the inorganic-organic parts. Although CPs can have many other potentials and fascinating properties, including heterogeneous catalysis, gas storage, gas separation, and drug carriers [11,12,13,14,15,16,17,18], magnetism is also an important area of interest [19,20,21,22,23,24,25,26,27,28,29]. This is particularly true, when the paramagnetic metal centers are bridged by short ligands (such as azido anion, cyanide, and carboxylate groups) to produce extended structures, 1D chains, and 2D layers, which is the structural basis for transmitting significant magnetic interactions between spin carriers and the metal ions [30,31,32]. One main benefit of magnetic coordination polymers (MCPs) is they provide the possibility and opportunity for tuning the nature of magnetic interactions within the materials [33,34,35,36]. Magnetically coupled interactions, ferromagnetic (FO), or antiferromagnetic (AF), depend on the types of spin carriers and a detailed understanding of the coupling pathway between them. The system and topology of magnetic interactions through space can be further adjusted if the starting metal centers, short bridging ligands, building blocks, co-ligands, templates, and related structures are carefully selected and accurately assembled into CPs [37,38,39].

Moreover, some MCPs can exist in bistable states, which could result in their structures being altered in response to external stimulation such as exchanging counterions, guest molecules, an electric field, or pressure. While in this dynamic structural transformation, their magnetic properties may obviously change. It is well-known that structural transformations can be observed in single-crystal-to-single-crystal transformations [40,41,42,43,44], as well as in crystal reassembly [35]. It follows that studies of magnetic materials and structural transformations of network materials could open opportunities for gaining an improved understanding of the essential characteristics that affect crystal nucleation and growth and thus would be of benefit for exploiting multi-functional materials.

One of the important characteristics for a polycarboxylate bridging ligand with metal centers is that they can give rise to a wide variety of multinuclear cluster-based compounds ranging from discrete units to multidimensional systems. Dimethylmalonic acid (H_2_-Me_2_mal) is one such potential candidate for the synthesis of CPs with diverting magnetic behaviors, such as ferromagnetism and magnetic ordering, because they not only link the metal centers to the multi-dimensional network but also reduce the distance between the metal centers, which would lead to a significant magnetic interaction.

In attempt to control reaction temperatures in solvothermal processes, we report herein on the synthesis of two Co(II) coordination polymers, [Co(Me_2_mal)(bpe)_0.5_(H_2_O)]*_n_* (**1**) and [Co(Me_2_mal)(bpe)_0.5_]*_n_* (**2**), from the self-assembly of H_2_-Me_2_mal and *trans*-1,2-bis(4-pyridyl)ethylene (bpe) in DMF/H_2_O. The temperature-induced network reassembly from compound **1** to compound **2** was also observed. Magnetic studies showed that complex **1** exhibits both spin-orbital coupling and antiferromagnetic interactions through the *syn*–*anti* carboxylate (Me_2_mal^2−^) bridge exchange pathway with a long Co–Co separation. In contrast, complex **2** shows a ferromagnetic interaction that occurs by an exchange pathway with a short Co–Co separation (3.150 Å) and small Co–O–Co bridging angles (98.5° and 95.3°) provided by the antisymmetric µ_4_-Me_2_mal^2−^ bridging ligand.

## 2. Experimental

### 2.1. Materials and Methods

All reagents and solvents were used as received without further purification, and aerobic conditions were performed for all reactions.

### 2.2. The Synthesis of [Co(Me_2_mal)(bpe)_0.5_(H_2_O)]_n_
***(1)***

A mixture of the solution of Co(CH_3_COO)_2_·4H_2_O (12.5 mg, 0.05 mmol), H_2_-Me_2_mal (6.6 mg, 0.05 mmol), bpe (9.1 mg, 0.05 mmol) and DMF/H_2_O (1 mL/5 mL) was heated at 80 °C for two days. After the mixtures were slowly cooled to room temperature, purple crystals of compound **1** suitable for single crystal X-ray analysis were obtained. The purple crystals were washed with water and collected by suction filtration. The simulated powder X-ray diffraction pattern from the single-crystal data was compared well with the pattern of the bulk sample (vide infra). The yield was 41% based on Co. Elemental analysis calcd (%) for C_11_H_13_CoNO_5_ (**1**): C, 44.27; H, 4.36; N, 4.70. Found: C, 44.23; H, 4.18; N, 4.60. IR data (KBr disk, cm^−1^): 3420 (s), 2979 (m), 2931 (w), 1610 (vs), 1534 (vs), 1460 (m), 1428 (s), 1376 (m), 1350 (w), 1254 (w), 1219 (w), 1192 (w), 1101 (w), 1071 (w), 1015 (m), 977 (w), 911 (w), 846 (m), 832 (m), 791 (m), 742 (m), 688 (w), 603 (w), 553 (m).

### 2.3. The Synthesis of [Co(Me_2_mal)(bpe)_0.5_]_n_
***(2)***

Method A: Co(CH_3_COO)_2_·4H_2_O (12.5 mg, 0.05 mmol), H_2_-Me_2_mal (6.8 mg, 0.11 mmol), dpe (9.1 mg, 0.05 mmol) and DMF/H_2_O (1 mL/5 mL) were mixed and placed in a Teflon reactor (25 mL). This mixture was heated to 120 °C at the heating rate of 14.3 °C/h, holding at 120 °C for 48 h, and then cooling to 30 °C at a rate of 1 °C/h. The red crystals of compound **2** suitable for single-crystal X-ray analysis were obtained. The crystals were washed a few times with water, collected by suction filtration and dried in air. The yield was 34% (based on Co). The simulated powder X-ray diffraction pattern from the single-crystal data was compared well with the pattern of the bulk sample (vide infra). Elemental analysis calcd (%) for C_11_H_11_CoNO_4_ (**2**): C, 47.12; H, 3.93; N, 5.00. Found: C, 47.25; H, 3.60; N, 5.04. IR data (KBr disk, cm^−1^): 3452 (s), 3071 (m), 2987 (m), 2957 (m), 2939 (m), 2864 (m), 1656 (vs), 1612 (vs), 1575 (vs), 1466 (m), 1433 (m), 1394 (vs), 1369 (s), 1345 (s), 1276 (s), 1196 (m), 1096 (w), 1078 (w), 1022 (m), 985 (m), 961 (w), 940 (w), 893 (m), 828 (m), 794 (w), 758 (w), 701 (m), 596 (m), 551 (m), 529 (w), 472 (w).

Method B: The crystalline samples of **1** (50.0 mg, 0.17 mmol) and DMF/H_2_O (1 mL/5 mL) were placed in a Teflon reactor (25 mL). The reactor was then heated to 120 °C at the heating rate of 14.3 °C/h, holding at 120 °C for 48 h, and then cooling to 30 °C at a rate of 1 °C/h. The red crystals were obtained, washed by water, dried in air, and collected by filtration. The yield was 77% (36.1 mg). The IR spectrum and powder X-ray diffraction pattern were identical to that of **2** prepared by method A. The elemental analysis calcd (%) for C_11_H_11_CoNO_4_ (**2**): C, 47.12; H, 3.93; N, 5.00. Found: C, 47.18; H, 3.69; N, 4.98.

### 2.4. X-ray Crystallography

For compounds **1** and **2**, the diffraction intensity data were collected at 150 K on a Bruker APEXII CCD diffractometer (Bruker, Karlsruhe, Germany) with graphite-monochromated Mo Kα radiation (λ = 0.7107 Å). The program SADABS (Bruker, 2016) was used for absorption corrections [45]. Direct methods were used to solve the structure, and the SHELX2014 program [46] was used to refine the structure with the full-matrix least-squares method against F^2^. All non-hydrogen atoms were refined anisotropic thermal parameters, whereas the hydrogen atoms on their respective carbon atoms were placed in ideal, calculated positions, using the riding model with isotropic thermal parameters. For compounds **1** and **2**, the experimental details for X-ray crystallographic data, and the refinements are summarized in Table 1, and the selected bond distances and angles are listed in Table 2 and Table 3.

### 2.5. Physical Measurements

The temperature dependence (dc) of the magnetic susceptibility measurements for compounds **1** and **2** were performed on microcrystalline samples, which were restrained in eicosane to prevent torquing, on a Quantum Design MPMS-7 SQUID (Quantum Design, San Diego, CA, USA) equipped with 7.0 T magnets and operated in the range of 2.0–300.0 K. The Pascal’s constants [47] were used to estimate the diamagnetic corrections of both compounds from the experimental magnetic susceptibilities to achieve the molar paramagnetic susceptibilities. Elemental analysis (carbon, hydrogen, and nitrogen) of compounds **1** and **2** were made using an Elemental vario EL III analyzer (PerkinElmer, Taipei, Taiwan). Thermogravimetric (TG) analyses of both compounds were collected using a Seiko Instrumental, Inc., (Chiba shi, Japan) EXSTAR 6200 TG/DTA analyzer, operating under a 5 °C/min heating rate and a nitrogen atmosphere. Powder X-ray diffraction data were collected using a Rigaku MiniFlex-II X-Ray diffractometer (Tokyo, Japan), operating on a step mode, with a step size of 0.02° in θ and a fixed time of 10 s at 40 kV, 30 mA for Cu–Kα (λ = 1.5406 Å). A Perkin-Elmer Spectrum RX1 FTIR spectrometer was used to collect the Fourier transform infrared (FTIR) spectra for both compounds (PerkinElmer, Taipei, Taiwan).

## 3. Results and Discussion

### 3.1. Synthesis and Characterization of Compounds ***1*** and ***2***

Compounds **1** and **2** were prepared via self-assembly processes by solvothermal reactions at different temperatures. The reaction of Co(OAc)_2_·4H_2_O (0.05 mmol) with H_2_-Me_2_mal (0.05 mmol) and bpe (0.05 mmol) in a DMF/H_2_O (1 mL/5 mL) mixed solution at 60 or 80 °C for 48 h yielded needle-shaped purple crystals of 1. No crystals of **1** were formed when the reaction was carried out at room temperature. When the above reaction was performed at 120 °C for 48 h, however, red crystals of thermodynamically stable products of compound 2 were obtained. When crystals of 1 in the same solvent system were allowed to stand at 120 °C for about 48 h, the purple colored crystals of **1** were converted into red colored crystals of 2, thus confirming that crystal reassembly has occurred. The synthesis and structural conversion of 1 to 2 are depicted in Scheme 1. However, except for **1,** no new product was formed, as determined by the PXRD patterns, but if the reaction was carried out in the absence of H_2_O/DMF as the solvent, this indicated that reassembly from 1 to 2 would be simultaneously assisted by both thermal and solvent factors. In viewing the crystal structures of **1** and **2**, the removal of coordinated water from the Co(II) center and the rearrangement of the carboxylate of Me_2_mal^2−^ and bpe ligands would be expected to take place during the structural conversion process. We assume that the solvent molecules would provide intermolecular interactions, i.e., hydrogen bonding interactions that would be expected to stabilize the structure in the intermediate state and would benefit from the removal of the coordinated water molecules. Such solvent assisted structural transformations have been reported in the literature [48,49,50,51], and these findings can be compared to the reassembly of crystals from **1** to **2**. In the reported literature, supramolecular interactions, such as layer-guest-layer hydrogen bonding interactions, stabilize the layer structure. As the temperature increased, the guest water molecules were removed and the structure was induced to transform into a 3D network. We, therefore, assume that the removal of coordination water molecules in **1** would be an important factor for the crystal reassembly to **2**.

The phase purity for the bulk samples of **1**, **2,** and the products obtained by the crystal reassembly of **2** was confirmed by PXRD (Appendix A) and by elemental analysis. The thermal stabilities of compounds **1** and **2** were also examined by TG analysis (Appendix A). The TGA curve for compound **1** showed consecutive weight loss steps at temperatures above 120 °C, which corresponded to the gradual removal of one coordinated water molecule (found: 6.28%; calcd: 6.04%). At temperatures above 250 °C, the host framework underwent a rapid weight loss, which was attributed to the elimination of the organic ligands and the gradual decomposition of the compound. Compound **2** showed one large weight loss at temperatures above 300 °C, corresponding to the elimination of the organic ligands, followed by the decomposition of the host framework.

### 3.2. Description of the Structure

#### Crystal Structures of Compound **1**

**[Co(Me_2_mal)(bpe)_0.5_(H_2_O)]*_n_* (1).** An X-ray structural analysis showed that compound **1** crystallized in the monoclinic space group *P*2_1_/*n* and the asymmetric unit of **1** contains one crystallographically independent Co(II) center, one Me_2_mal^2−^ anion, one-half of a bpe ligand, and one coordinated water molecule. As depicted in Figure 1, the geometry of the Co(II) center is a distorted octahedron with a CoO_5_N coordination sphere. The equatorial positions on the octahedron are occupied by four oxygen atoms (O2, O3, and their symmetrical equivalents) derived from three Me_2_mal^2−^ ligands, while one nitrogen atom (N1) from one bpe ligand and one oxygen atom (O6) of a terminal water molecule occupied the axial positions. The Co–O bond lengths at the Co(II) vary from 2.0747(18)–2.124(19) Å and a Co–N bond length is 2.151(2) Å, which falls in the range of values for typical octahedral Co(II) complexes [52,53]. The two carboxylate groups of the Me_2_mal^2−^ ligand adopt a *syn*–*anti* µ_2_:η^1^,η^1^-bridging mode (Scheme 2a), in which the Co(II) ion is chelated by each of one oxygen atoms (O2 and O3), and connects to two crystallographically equivalent Co(II) ions through each of the other two oxygen atoms (O1 and O4). Thus, each Co(II) center is linked to four neighbors via three Me_2_mal^2−^ bridges and results in a corrugated Co-Me_2_mal layer that is located parallel to the *a**c* crystal plane (Figure 2a). The two unique Co···Co distances in the layer spanned by the Me_2_mal^2−^ ligands are 5.478(1) and 5.258(1) Å. The adjacent Co–Me_2_mal layers are further pillared through *bis*-monodentate bpe ligands leading to a pillared-layer 3D framework with 1D channels along the crystallographic *c* axis (Figure 2b). Similar structures have been reported in previous studies [54,55,56]. The bpe ligands are positioned alternately above and below the layers, in a *trans* array with the dimethyl groups of the Me_2_mal^2−^ ligand. The bridging bpe ligands separate the Co(II) ions by 13.668(2) Å, where the shortest interlayer Co···Co distance is 9.911(2) Å. The shortest distance of the centroid–centroid between the adjacent pyridyl rings of the bpe is 7.496(2) Å, which rules out any *π*–*π* interaction in compound **1**. The shortest centroid–centroid distance between the adjacent pyridyl rings of the bpe ligand in compound **1** is 7.496(2) Å, which are considerably higher than the limit for *π*–*π* interactions between pyridyl rings, thus indicating there are no *π*–*π* interactions in compound **1**. The intralayer hydrogen bonds between the oxygen atoms of the Me_2_mal^2-^ ligand and the coordinated water molecule in **1** (2.704(3) and 2.768(3) A° for O6···O2 and O6···O3 and 150.4(2) and 163.9(4)^o^ for O6–H6A…O2 and O6–H6B···O4) donate to the stabilization of the structure.

Topological analysis of the 3D structure of compound **1** indicated that each Co(II) center with a CoNO_5_ coordination environment and µ_3_-Me_2_mal can be viewed as a six-connected node and a three-connected node, respectively, while each bpe ligand that is bonded to two Co(II) centers can be treated as a two-connector. Such connectivity repeats in infinity, thus producing the Co-Me_2_mal layer and the 3D framework of **1** as schematically represented in Figure 3. Analysis by the TOPOS software package showed that the framework of **1** could be explained as a binodal (3,4)-connected **ins** topology with the Schläfli symbol (6^3^)(6^5^.8) [57].

**[Co(Me_2_mal)(bpe)_0.5_]*_n_* (2)**. A single-crystal X-ray diffraction analysis reveals that **2** is a three-dimensional framework consisting of well-isolated one-dimensional cobalt/oxygen chains bridged by a bpe ligand along with a Me_2_mal ligand. The asymmetric unit of compound **2** contains two Co(II) atoms at special positions, two Me_2_mal^2−^ groups, and one bpe ligand (Figure 4). The Co1 shows a distorted octahedral with a CoO_4_N_2_ coordination sphere, where two oxygen atoms (O1, O1A) of the carboxylate from the two Me_2_mal^2−^ ligands and two nitrogen atoms (N1, N1A) from the two bpe ligands make up the equatorial plane, and the two carboxylate oxygen atoms (O3, O3A) from two Me_2_mal^2−^ ligands occupy the axial positions. The Co2 center adopts a CoO_6_ distorted octahedral geometry bonded to six carboxylate oxygen atoms from four different Me_2_mal^2−^ ligands, where the four oxygen atoms (O1, O2, and their symmetric equivalents) occupy the equatorial plane and the two oxygen atoms, O3 and its symmetric equivalent, are coordinated to the axial positions. The Co−O bond distances (ranging from 1.992(2)–2.163(2) Å) and Co−N (2.067(3) Å) are all within the normal ranges for octahedral Co(II) complexes [42,43]. The Me_2_mal^2−^ ligand acts as an asymmetrical µ_4_-bridging ligand (Scheme 2b). One of two carboxylates in a Me_2_mal^2−^ adopt a *anti*, *syn*–*syn,* µ_3_:η^2^,η^1^-bridging mode with terminal bonding to the Co2A center through an O2 atom with the Co1 and Co2 centers bridged by the O2 atom, while the other one of two carboxylate groups show an *anti*,*syn* µ_2_:η^2^-bridging mode that connects the Co2 to Co1A centers through its one carboxylate oxygen atom (O3) with an uncoordinated oxygen atom (O4). Such an asymmetrical µ_4_-bridging structure was first obtained in the complex with malonate-related ligands. Therefore, the Co1 and Co2 centers are connected by one *syn*–*syn* carboxylate and two µ_2_-oxygen atoms derived from two carboxylate groups and lead to the formation of an edged-shared zigzag Co−Me_2_mal chain along the *c* axis (Figure 5a). The intrachain Co1−Co2 bond distance is 3.150 Å, and the Co1−O1−Co2 and Co1−O3−Co2 bond angles are 95.35° and 98.51°, respectively. The adjacent Co−Me_2_mal chains are further crosslinked through *bis*-monodentate bpe ligands leading to a 3D network structure with 3D channels (Figure 5b). In the crystal, the adjacent porous 3D framework is interpenetrated leading to a two-fold interpenetrated network, in which only the small 1D channels along the *a* axis can be observed (Appendix A). In the 3D interpenetrated framework, the bridging bpe ligands separate the Co(II) ions by 13.287(4) Å, where the shortest interchain Co···Co distance is 9.352(2) Å. The shortest centroid–centroid distance between adjacent pyridyl rings of bpe is 7.496(2) Å ruling out any *π*–*π* interactions in compound **2**.

### 3.3. Magnetic Properties

Solid-state, temperature-dependence of magnetic susceptibility data of compounds **1** and **2** were measured in the 2.0–300 K range under a 1.0 kOe applied field.

The temperature dependence of χ_M_*T* values of compound **1** is shown in Figure 6. The spin-only value at 300 K is 3.34 cm^3^ K mol^–1^, which is in agreement with that of the measured, calculated by a high-spin Co(II) ion with *S* = 3/2 with a strong spin-orbit coupling [58]. The χ_M_*T* value of **1** continues to decrease with cooling of the temperature from 300 to 10 K, below 10 K the value of χ_M_*T* decreases more rapidly to 1.40 cm^3^ K mol^–1^ at 2.0 K. The monotonic decrease in the χ_M_*T* value with decreasing temperatures are characteristics of overall antiferromagnetic interactions and/or spin-orbital couplings in compound **1**. Above 50 K, the magnetic susceptibility data obeyed the Curie–Weiss law with a Curie constant *C* = 3.58 cm^3^ mol^–1^ K and Weiss *θ* = −23.15 K (Appendix A). The negative *θ* value indicates that antiferromagnetic coupling and the spin-orbital coupling existed in compound **1**. 

A plot of the temperature dependence of the χ_M_*T* value for compound **2** is shown in Figure 7. The χ_M_*T* value is 3.13 cm^3^ K mol^−1^ at 300 K which is significantly greater than the spin-only value for a the high-spin Co(II) center, while it is in agreement with the values observed of the magnetic moment for high-spin Co(II) complexes in an octahedral environment with strong spin-orbital coupling. The χ_M_*T* value decreases slowly to a minimum of 3.03 cm^3^ K mol^−1^ at 55 K. Below 55 K, the χ_M_*T* value sharply increases to reach a maximum of 9.12 cm^3^ K mol^−1^ at 2.0 K, indicating the existence of the ferromagnetic interaction in compound **2**. The data of the temperature dependence of magnetic susceptibilities above 50 K followed the Curie−Weiss law giving a θ = −2.90 K and a *C* = 3.21 cm^3^ K mol^−1^ (Appendix A). The Curie constant of **2** was larger than the theoretical spin-only value of the Co(II) ion, indicating that an orbital contribution existed in compound **2**. Thereby, the weak and negative values of θ are unable to indicate the antiferromagnetic interaction between Co(II) centers in **2** because of the significant strong spin-orbital coupling of the octahedral Co(II) ions.

Because of the contribution of the spin-orbit coupling for Co(II) ions, it is not possible to find a suitable analytical expression that describes the temperature-dependent magnetic susceptibility for Co(II) centers of the layered and chained polymeric structures in compounds **1** and **2**, respectively. However, Rueff et al. successfully proposed a phenomenological approach for a low-dimensional polymeric Co(II) compound that permits the magnitude of the magnetic coupling and the spin-orbit coupling effects. They assumed the phenomenological equation [59]:χ_M_*T* = *A* exp(−*E*_1_/*k*T) + *B* exp(−*E*_2_/*kT*) where *A* + *B* are the Curie constants and the *E*_1_ and *E*_2_ are the activation energies, which correspond to the parameters of the spin–orbit coupling and the magnetic coupling interaction, respectively. The *E*_2_ is related to the constant of the magnetic coupling (*J*) according to the Ising chain approximation, χ_M_*T* ∞ exp(+*J*/2*kT*). This equation sufficiently pronounces the spin-orbit coupling, which affects the splitting of the low-temperature divergence of the susceptibility between discrete levels and the exponential. Some reasonable values for magnetic interactions and interactions of spin–orbit coupling have been described in several studies on Co(II) coordination polymers with 1D and 2D structures [60,61,62]. 

The results obtained by the Rueff′s procedure are quite consistent with the experimental data. For compound **1**, the data above 10 K was fitted and the parameters of the fitting are *A* + *B* = 3.72 cm^3^ K mol^–1^, practically close to the Curie constant found from the Curie–Weiss law, 3.58 cm^3^ K mol^–1^. The *E*_1_/*k* was +49.73 K which is in the same magnitude (the order of +100 K) to those reported for Co(II) compounds. Concerning the values obtained for antiferromagnetic exchange interactions, it is weak but significant (*E*_2_/*k* = 0.46 K, corresponding to *J* = −0.92 K), which is in agreement with the antiferromagnetic property of compound **1** and consistent with some other reported 2D Co(II) compounds [61,62]. As described in the crystallographic part, compound **1** is comprised of Co(II) ions connected by carboxylate groups in a *syn*–*anti* bridging mode thus giving a Co-Me_2_mal layer, which is further linked in a three-dimensional network by bpe ligands. Thus, the overall antiferromagnetic exchange interaction can be attributed to magnetic interaction within the Co–carboxylate layer. Magnetic exchanges through the *syn*–*anti* carboxylate bridges for Co(II) and Mn(II) ions are usually reported as antiferromagnetic due to the good overlap of magnetic orbitals [54,55,56]. For compound **2**, the data above 30 K were fitted and the best fit parameters of the *A* + *B* value was 3.18 cm^3^ K mol^–1^, which is in good agreement with those reported in the literature for the Curie constant, *E*_1_/k, the effect of the distortion of coordination site and spin-orbit coupling, was +99.41 K and −*E*_2_/k was 4.07 K, corresponding to magnetic interactions of *J* = 8.14 K within the Ising chain approximation, which is consistent with values reported for several 1D Co(II) complexes [59,60,61]. These fitting results indicate that the distinct ferromagnetic exchange is dominated between Co(II) through one O−C−O and two µ_2_-O bridges. The intrachain Co1−O−Co2 angles and Co···Co distance are 98.5°, 95.3° and 3.150(1) Å, respectively. Nevertheless, the shortest Co···Co distance in the chain is 9.352 Å. The magnetic interactions of compounds **1** and **2** can be compared to those of Co(II) compounds containing similar structures in the literature [60,63], in which a weak antiferromagnetic interaction (*J* = −1.30 K) was dominated in a *syn*–*anti* carboxylate-bridged Co(II)-malonate layer and a weak ferromagnetic coupling (*J* = 3.72 K) was transmitted in a Co(II)-based chain with one O−C−O and two µ_2_-O bridges. This is in agreement with magneto-structural analyses: a small Co−O−Co angle resulting in a Co−Co ferromagnetic coupling and a large Co−O−Co angle resulting in a Co−Co antiferromagnetic interaction [63].

The ferromagnetic coupling of compound **2** was further estimated by isothermal magnetization data. As shown in Appendix A, the magnetization increased sharply and then reached a saturation plateau (2.53 Nβ at 7 T) with a fast saturation of the magnetization. The fast saturated magnetization confirms the existence of ferromagnetic interactions within **2** and the saturation value was consistent with the theoretical values (2−3 *N*β) expected for Co(II) compounds. Indeed, no magnetic hysteresis loop was detected indicating the absence of magnetic ordering in **2** above 2.0 K.

## 4. Conclusions

In summary, we report on the temperature-controlled synthesis and characterization of two new dicarboxylate-based 3D Co(II) coordination polymers. The formation of compound **1** was favored at lower temperatures of 60–80 °C, but a higher temperature of 120 °C was favored to yield compound **2**. Indeed, a crystal reassembly from compound **1** to compound **2** was also observed by solvothermal treatment of **1** at 120 °C in DMF/H_2_O. The structure of compound **1** contains corrugated layers of Co(II) connected through *syn*–*anti* carboxylate bridges of Me_2_mal^−^ ligands and *bis*-monodentate bpe pillars. Compound **2** shows a 3D porous framework involving one-dimensional Co–carboxylate chains connected by antisymmetric µ_4_-Me_2_mal^2−^ and *bis*-monodentate bpe ligands. Magnetic measurements indicate that antiferromagnetic interactions through the *syn*–*anti* carboxylate bridges of the Me_2_mal^2−^ ligands were dominated in compound **1**, while compound **2** revealed ferromagnetic interactions resulting from the short Co–Co separation (3.150 Å) and small Co–O–Co bridging angles (98.5° and 95.3°) exchange pathway of the µ_4_-Me_2_mal^2−^ bridges. The studies demonstrate that dimethylmalonic acid has great substantial for use in the preparation of coordination polymers with multipurpose structural topologies and unusual magnetic properties.

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
