# Peer review of "Temperature-Controlled Assembly/Reassembly of Two Dicarboxylate-Based Three-Dimensional Co(II) Coordination Polymers with an Antiferromagnetic Metallic Layer and a Ferromagnetic Metallic Chain"

_polymers, 2019, doi:10.3390/polym11050795_

Round 1
Reviewer 1 Report
Yang and coworkers report on the synthesis and characterization of two Co(II)-based coordination polymers through a temperature-controlled process. The CPs are well characterized by X-ray single crystal diffraction, and interestingly they exhibit different magnetic properties, antiferromagnetic and ferromegnetic, although they have similar chemical compositions. The following issues are provided for the authors to consider in revision.
To emphasize the temperature-controlled assembly, it is suggested that more temperature points should be examined instead of only showing two points. In addition, in the experimental section, compound 1 was prepared at 60 oC, but in Scheme 1 and main text, it says 80 oC. The temperatures should be consistent.
Many grammatical errors and typos are found. The English writing should be improved. Some errors are listed as follows: a) Line 65: "4,4'-bipyridinly-ethylene" should be read as "trans-1,2-bis(4-pyridyl)ethylene". b) Line 78: "dpe" should be changed to "bpe". c) Line 300: "seasonable" => "reasonable"? d) "ml" => "mL".
Author Response
1. The reaction of compound 1 was operated at 60 oC and 40 oC as reviewer’s suggestion and these experimental results have been added to manuscript (L 147-149).
2. L70, 83, 94, 95, 105, 106 and 318. The typos have been addressed as referee’s suggestion.
Reviewer 2 Report
Yang et al reported two dicarboxylate-based three-dimensional Co(II) coordination polymers showing an antiferromagnetic metallic layer and a ferromagnetic metallic chain. More interesting, 1 can be transformed to 2 in DMF/H2O in higher reaction temperature. This work is well organized and characterized fully in consistent with the conclusions. The presentation of the data is expertly done and the manuscript is written very concisely. I would like to recommend acceptance of this manuscript after minor revisions (see my detailed comments below).
(1) The reported two compounds are coordination polymers, so the formula for them should be ended with n after the square brackets.
(2) Many special symbols for coordination modes were shown incorrectly in the pdf version of manuscript.
(3) The word of “poly-carboxylate” looks not suitable and should be revised to dicarboxylate.
(4) Some related references should be noted such as Chinese Chem Lett 2014, 25, 835, Inorg. Chem., 58, (2019) 4574; Chem. Eur. J. 23, (2017) 7990; Inorg. Chem. 2016, 55, 5139, Inorganic Chemistry Frontiers 2018, 5, 2314, Dalton Trans., 2017, 46, 2137.
Author Response
L12, 68, 69, 81, 92, 191, and 239. The “n” has been added to the end of formula as referee’s suggestion
The incorrectly symbols have been addressed.
L3, 11 and 358. The “poly-carboxylate” has been changed to “dicarboxylate” as referee’s suggestion.
The references have been added and cited in refs. 24-29 as referee’s suggestion.
Reviewer 3 Report
This manuscript reports the synthesis, characterization and magnetic properties of two 3D coordination polymers. The work is systematically planned and the observed results are neatly discussed. This is a nice contribution and hence this work may be considered for publication in Polymers. However, the following minor changes may be addressed before final acceptance.
· Authors may comment on various applications of CPs including heterogeneous catalysis, gas storage, gas separation, drug carriers etc. Some of the recent reviews may be added (Chem. Soc. Rev., 2018,47, 8134-8172; ACS Catal., 2019, 9 (2), pp 1081–1102; ChemCatChem, 2019, 11, 899-923)
· The observed properties of these Co CPs may be compared with the literature data.
Author Response
The references have been added and cited in refs. 16-18 as referee’s suggestion.
L164-169, L342-346. The comparisons of the observed properties for compounds 1 and 2 have been added to the manuscript as reviewer’s suggestion.